# Altered Mitochondrial Dynamic in Lymphoblasts and Fibroblasts Mutated for *FANCA-A* Gene: The Central Role of DRP1

**DOI:** 10.3390/ijms24076557

**Published:** 2023-03-31

**Authors:** Nadia Bertola, Silvia Bruno, Cristina Capanni, Marta Columbaro, Andrea Nicola Mazzarello, Fabio Corsolini, Stefano Regis, Paolo Degan, Enrico Cappelli, Silvia Ravera

**Affiliations:** 1Department of Experimental Medicine, University of Genoa, Via De Toni 14, 16132 Genova, Italy; 2CNR Institute of Molecular Genetics “Luigi Luca Cavalli-Sforza”, Unit of Bologna, Via di Barbiano 1/10, 40136 Bologna, Italy; 3IRCCS Rizzoli Orthopedic Institute, Via Pupilli 1, 40136 Bologna, Italy; 4Electron Microscopy Platform, IRCCS Istituto Ortopedico Rizzoli, 40136 Bologna, Italy; 5Laboratory for the Study of Inborn Errors of Metabolism—Pediatric Clinic and Endocrinology IRCCS Istituto Giannina Gaslini, Via Gerolamo Gaslini 5, 16148 Genova, Italy; 6Laboratory of Clinical and Experimental Immunology, IRCCS Istituto Giannina Gaslini, 16147 Genova, Italy; 7U.O. Mutagenesis, IRCCS AOU San Martino—IST (Istituto Nazionale per la Ricerca sul Cancro), Largo Rosanna Benzi 10, 16132 Genova, Italy; 8Hematology Unit, IRCCS Istituto Giannina Gaslini, Via Gerolamo Gaslini 5, 16148 Genova, Italy

**Keywords:** OxPhos, aerobic metabolism, fusion, fission, mitochondrial dynamics, Fanconi anemia

## Abstract

Fanconi anemia (FA) is a rare genetic disorder characterized by bone marrow failure and aplastic anemia. So far, 23 genes are involved in this pathology, and their mutations lead to a defect in DNA repair. In recent years, it has been observed that FA cells also display mitochondrial metabolism defects, causing an accumulation of intracellular lipids and oxidative damage. However, the molecular mechanisms involved in the metabolic alterations have not yet been elucidated. In this work, by using lymphoblasts and fibroblasts mutated for the *FANC-A* gene, oxidative phosphorylation (OxPhos) and mitochondria dynamics markers expression was analyzed. Results show that the metabolic defect does not depend on an altered expression of the proteins involved in OxPhos. However, FA cells are characterized by increased uncoupling protein UCP2 expression. *FANC-A* mutation is also associated with DRP1 overexpression that causes an imbalance in the mitochondrial dynamic toward fission and lower expression of Parkin and Beclin1. Treatment with P110, a specific inhibitor of DRP1, shows a partial mitochondrial function recovery and the decrement of DRP1 and UCP2 expression, suggesting a pivotal role of the mitochondrial dynamics in the etiopathology of Fanconi anemia.

## 1. Introduction

Fanconi anemia (FA) is a rare genetic disorder, recessive autosomal or X-linked [1]. FA phenotype can be very heterogeneous, with clinical manifestations ranging from congenital malformations to metabolic dysfunction susceptibility and increased risk of cancer developing, particularly leukemia and squamous cell carcinoma [2,3]. However, bone marrow failure and aplastic anemia are the most common causes of death in FA patients, which develop at a younger age and with a 5000-fold increased risk compared to the healthy population [3]. So far, 23 genes have been identified as involved in FA, with 90% of mutations occurring in *FANC-A*, *FANC-C*, or *FANC-G* genes [4], where *FANC-A* represents accounts for two-thirds of cases. *FANC* genes encode proteins that assemble into a complex involved in DNA damage repair, specifically interstrand cross-links [5,6]. Recently, FA pathogenesis has also been associated with mitochondrial defects, which cause a metabolic shift toward anaerobic respiration, lipid accumulation, and unbalanced oxidative stress [7,8,9]. Specifically, the electron transport between respiratory complexes I and III is impaired [10], triggering elevated reactive oxygen species production (ROS) not counteracted by cellular antioxidant defenses [11,12]. These metabolic dysfunctions are associated with altered mitochondrial morphology, as mitochondria appear more swollen and with less defined cristae [10,13,14,15,16]. Although literature confirms the metabolic dysfunction in FA [17,18,19,20], the molecular mechanisms correlating *FANC* genes’ mutations with these defects are still to be clarified. Since the mitochondrial metabolism extent and its efficiency are strictly related to the mitochondrial network shape [21], several authors have described a defect in the mitochondrial dynamic, autophagy, and mitophagy processes [22,23,24,25]. 

Thus, this work aims to deeply investigate whether the mitochondrial metabolism alterations observed in FA depend on altered expression of proteins belonging to the oxidative phosphorylation (OxPhos) machinery or mitochondrial biogenesis and dynamics modulators in lymphoblasts and fibroblasts carrying the mutated *FANC-A* gene, comparing the results with isogenic-corrected *FANC-A* gene cell lines. 

Results show that cells carrying the *FANC-A* mutation exhibit an overexpression of UCP2 and DRP1, suggesting that the altered mitochondria metabolism could depend on oxidative phosphorylation uncoupling and an imbalance toward mitochondrial fission. Treatment with P110, a specific inhibitor of DRP1 [26], can partially reverse the metabolic dysfunction and the organization of the mitochondrial network. Furthermore, FA cells show lower expression of Beclin1 and Parkin.

## 2. Results

### 2.1. FANC-A Lymphoblasts and Fibroblasts Display Damaged Mitochondria Unable to Conduct an Efficient OxPhos

As previously reported in the literature [7,10,12,15,18], lymphoblasts and fibroblasts carrying the *FANC-A* gene mutation are characterized by an altered pyruvate/malate-induced oxygen consumption rate (OCR), partially compensated by respiration led by the complex II pathway (Figure 1A–C and Figure 2A–C). The dysfunctional respiration is associated with the ATP synthesis decrement (Figure 1D and Figure 2D) and a reduced OxPhos efficiency, as shown by the P/O values (Figure 1E and Figure 2E). The impaired OxPhos function depends on an altered electron transfer between respiratory complexes I and III (Figure 1F and Figure 2F). In addition, electron microscopy analysis on FA lymphoblasts shows swollen mitochondria with a disorganized inner membrane (Figure 1G and Figure 2G). 

### 2.2. Dysfunctional Mitochondria Metabolism in FANC-A Cells Does Not Depend on OxPhos Protein Expression Alteration but Appears Correlated with Increased UCP2 Expression

To understand whether defective OxPhos in FA cells depends on an altered expression of respiratory chain proteins, Western blot analyses were performed against ND1 (a Complex I subunit, mitochondrial DNA-encoded), SDHB (a Complex II subunit, nuclear DNA-encoded), MTCO2 (a Complex IV subunit, mitochondrial DNA-encoded), and β subunit of ATP synthase (a subunit of F_1_ moiety nuclear-DNA encoded) in lymphoblast (Figure 3A) and fibroblast (Figure 3B) cell lines. Data do not show significant differences in the expression of these proteins in both FA cell models compared to FAcorr. By contrast, *FANC-A* lymphoblast and fibroblast cell lines display a significant increase in uncoupling protein 2 (UCP2) expression in *FANC-A* mutated cells compared to their respective controls (Figure 3A,B), explaining the lower P/O values in *FANC-A* cells reported in Figure 1E and Figure 2E. 

### 2.3. Mitochondrial Dynamic Is Unbalanced in FANC-A Cells

Mitochondrial biogenesis and dynamics play a pivotal role in the OxPhos efficiency main promoting the mitochondrial network organization maintenance [27]. Thus, the expression of proteins involved in mitochondrial biogenesis and fusion and fission processes was analyzed.

*FANC-A* lymphoblasts (Figure 4A) and fibroblasts (Figure 4B) do not show significant differences in the expression of CLUH, an mRNA-binding protein involved in mitochondrial biogenesis, and OPA1 and MFN2, two proteins involved in mitochondrial fusion, compared to the controls. Conversely, expression of DRP1, a protein involved in mitochondrial fission, appears higher in *FANC-A* cells than in FAcorr cells. This alteration suggests imbalance in mitochondrial dynamics, where mitochondrial fission is promoted over fusion, leading to mitochondrial network disruption (Figure 4C) and causing a consequent OxPhos efficiency decrease.

### 2.4. Overexpressed DRP1 Reduction Partially Restores the OxPhos Activity

Since an unbalanced dynamic toward fission disrupts the mitochondrial network and lowers the OxPhos energy efficiency [28], *FANC-A* lymphoblasts and fibroblasts were treated for 24 h with P110, a specific DRP1 inhibitor [26]. Following this treatment, the overexpression of DRP1 is reduced in both treated *FANC-A* cell models compared to untreated cells, approaching the expression levels of the corrected cells (Figure 5A for lymphoblasts and Figure 6A for fibroblasts). This reduction is associated with a recovery of mitochondrial network organization, as the balance between fusion and fission was partially restored (Figure 6C).

P110 treatment also causes an increase in complexes I-III electron transport chain as well as oxygen consumption and ATP synthesis stimulated by pyruvate/malate (Figure 5B for lymphoblasts and Figure 6B for fibroblasts), causing a recovery in OxPhos efficiency, probably due to both decreased UCP2 expression and improved mitochondrial reticulum organization. In addition, the energy metabolism amelioration results in fatty acid accumulation and lipid peroxidation reduction (Figure 5B for lymphoblasts and Figure 6B for fibroblasts).

### 2.5. FANC-A Cells Display a Lower Protein Expression of Parkin and Beclin1 

Since the literature reports that FA cells display dysfunctional mitophagy and autophagy [22,23,24,25], the expression of markers involved in these processes was evaluated. Data show that both in lymphoblasts (Figure 7A) and fibroblasts (Figure 7B), Parkin, an E3 ubiquitin ligase involved in the mitochondrion polyubiquitination [29], and Beclin1, an autophagy activator [30], are lower expressed than in FAcorr cell. Conversely, Pink1 and several autophagy effectors such as LC3, Atg7, Atg12, and Atg16L1 appear similar in *FANC-A* cells compared to the control. 

## 3. Discussion

*FANC-A* mutated cells show damaged mitochondria associated with an impaired OxPhos function due to altered electron transfer between complexes I and III, confirming data reported in the literature [7,10,12,18]. This metabolic alteration causes a reduction in the cellular energy status, lipid droplet accumulation, and an oxidative damage increment [12,16,31]. However, the mechanism that causes this alteration has not yet been elucidated. Therefore, considering that mitochondrial function depends on the OxPhos machinery integrity as well as the balance in the mitochondrial biogenesis and dynamics [32], the protein expression of some respiratory complexes subunits, mitochondrial fusion/fission, mitophagy, and autophagy markers were analyzed in lymphoblasts and fibroblasts mutated for *FANC-A*.

The expression evaluation of SDHB, a subunit of complex II, and beta subunit of ATP synthase, encoded by nuclear DNA [33,34], showed no significant differences between cells carrying the *FANC-A* mutation and the control cells. The same result was obtained by assessing the expression of ND1 and MTCO2, subunits of complex I and II, respectively, encoded by mitochondrial DNA [35]. Thus, it is possible to hypothesize that the metabolic defect does not depend directly on the altered expression of the OxPhos machinery, either at the nuclear or mitochondrial level. Nevertheless, FA cells show increased expression of UCP2, an uncoupling protein, which promotes the proton passage across the inner mitochondrial membrane, dissipating the proton gradient [36]. This increase could justify the uncoupling between ATP synthesis and oxygen consumption observed in FA cells, which causes oxidative stress increment and consequent DNA damage. UCP2 also regulates mitochondrial calcium uptake [37], and its overexpression could play a role in dysregulated calcium homeostasis observed in FA cells [38]. 

The OxPhos efficiency depends on the balance between fusion and fission processes [39], which regulate mitochondrial dynamics [40]. Fusion activation induces mitochondrial elongation and increased development of the mitochondrial network, promoting the OxPhos functionality and the interaction between mitochondria and other cellular organelles, including the endoplasmic reticulum [41]. Conversely, fission determines the breakdown of the mitochondrial network, a necessary condition during cell division [41]. However, isolated mitochondria appear much less efficient in energy production than those organized in a reticulum [42]. Based on the WB analysis shown in Figure 3, FA cells show similar expression of CLUH, an RNA-binding protein involved in mitochondria biogenesis, and OPA1 and MFN2, two proteins involved in fusion, but higher levels of DRP1, a GTPase involved in fission. Consequently, cells mutated for *FANC-A* display altered mitochondrial dynamics biased toward mitochondrial reticulum disaggregation, as shown by confocal microscopy images. On the other hand, literature and data reported in Figure 1 report that mitochondria in FA cells appear smaller and swollen with poorly defined ridges [10,13,14,15,16], all characteristics attributable to increased mitochondrial fission. On the contrary, the Parkin and Beclin1 expression appears lower than in FAcorr cells, data in line with the alteration in mitophagy and autophagy reported in the literature, and that may justify the damaged mitochondrial accumulation [22,23,24,25].

The pivotal role of DRP1 overexpression in the metabolic dysfunction in FA cells is confirmed by the partial restoration of the mitochondrial dynamic and functionality observed after treatment with P110, a specific DRP1 inhibitor [26]. In detail, cells treated with P110 show a DRP1 expression like the control and a better-organized mitochondrial network. In addition, treatment with P110 improves mitochondrial function and efficiency, as the electron transport between complexes I and III is partially restored, resulting in an amelioration of OCR and ATP synthesis through the pathway led by complex I. In addition, treatment with P110 also reduces the expression of UCP2, improving energy efficiency. The aerobic metabolism improvement is associated with a lower accumulation of fatty acids and subsequent lipid peroxidation, improving the cell’s overall redox state. On the other hand, Shyamsunder et al. have already suggested that DRP1 is involved in autophagy and mitophagy alteration in FA cells [22]. 

However, the mechanisms linking altered mitochondrial dynamics to FA gene mutations are still unclear. In this regard, Gueiderikh et al. recently demonstrated the involvement of FA protein in transcription as involved in ribosome biogenesis and nucleolar maintenance [43], which could explain the altered expression of proteins belonging to different pathways. Furthermore, an altered expression profile of miRNAs could also be involved [44].

Thus, although the link between the FA mutation and altered mitochondrial dynamics remains to be elucidated, this work suggests that the mitochondrial dynamic modulation plays a pivotal role in the pathogenesis of FA and that its restoration could be considered a therapeutic target.

## 4. Materials and Methods

### 4.1. Materials

All chemical compounds were of the highest chemical grade (i.e., Tris-HCl, KCl, EGTA, MgCl_2_, sulfuric acid, trichloroacetic acid, and HCl) and were purchased from Sigma-Aldrich, St. Louis, MO, USA.

### 4.2. Cellular Models and Treatment

Three different FANC-A lymphoblast cell lines (Lympho FA) and three different *FANC-A* primary fibroblast cell lines (Fibro FA) derived from four patients who carried out different mutations of FANC-A gene were obtained from the “Cell Line and DNA Biobank from Patients affected by Genetic Diseases” (G. Gaslini Institute)—Telethon Genetic Biobank Network (Project No. GTB07001) [10]. In addition, isogenic FA-corr cell lines, generated by the same *FANC-A* lymphoblast and fibroblast cell lines corrected with S11FAIN retrovirus (Lympho FAcorr and Fibro FAcorr), were employed as a control to maintain the characteristics of the FA cell lines except for the *FANC-A* gene mutation [10]. 

Fibroblast cell lines were grown as a monolayer at 37 °C with a 5% CO_2_ in DMEM high glucose with glutamax^®^ (#61965026, GIBCO, Billing, MT, USA), containing 10% fetal bovine serum (FBS; #ECS0120L, Euroclone, Milano, Italy), 100 U/mL penicillin, and 100 μg/mL streptomycin (#ECB3001D, Euroclone, Milano, Italy). Lymphoblast cell lines were grown in RPMI-1640 medium (#21875091, GIBCO, Billing, MT, USA) containing 10% FBS (#ECS0120L, Euroclone, Italy), 100 U/mL penicillin, and 100 μg/mL streptomycin (#ECB3001D, Euroclone, Milano, Italy) at 37 °C with a 5% CO_2_ [10]. All cell lines were provided by “Cell Line and DNA Biobank from patients affected by Genetic Diseases” (G. Gaslini Institute)—Telethon Genetic Biobank Network. 

In some experiments, lymphoblasts and fibroblasts were treated for 24 h with 1 µM P110 (#6897/1, Bio-Techne, Minneapolis, MN, USA), a specific inhibitor of DRP1, a protein involved in the mitochondrial fission process [26].

### 4.3. Oxygen Consumption Rate Assay 

Oxygen consumption rate (OCR) was measured with an amperometric electrode (Unisense Microrespiration, Unisense A/S, Aarhus, Denmark) in a closed chamber at 25 °C. For each experiment, 10^5^ cells permeabilized with 0.03 mg/mL digitonin for 1 min were employed. Additionally, 10 mM pyruvate plus 5 mM malate (#P4562 and #M8304, respectively, Sigma-Aldrich, St. Louis, MO, USA) were added to stimulate the respiratory pathway composed of Complexes I, III and IV; 20 mM succinate (#S7501, Sigma-Aldrich, USA) were added to stimulate the pathway composed of Complexes II, III, and IV [7,12]. As specific inhibitors of Complex I and Complex 3, 10 µM rotenone (#R8875, Sigma-Aldrich, St. Louis, MO, USA), or 50 µM antimycin A (#A8674, Sigma-Aldrich, St. Louis, MO, USA) were added, respectively.

### 4.4. F_o_F_1_ ATP-Synthase Activity Assay 

The F_o_F_1_ ATP-synthase activity was evaluated by incubating 10^5^ cells at 37 °C for 10 min. Incubation was done in a medium containing 50 mM Tris-HCl (pH 7.4), 50 mM KCl, 1 mM EGTA, 2 mM MgCl_2_, 0.6 mM ouabain (#O0200000, Sigma-Aldrich, St. Louis, MO, USA), 0.25 mM di(adenosine)-5-penta-phosphate (an adenylate kinase inhibitor, #D1387, Sigma-Aldrich, St. Louis, MO, USA)), and 25 μg/mL ampicillin (0.1 mL final volume, #A9393, Sigma-Aldrich, St. Louis, MO, USA). Additionally, 10 mM pyruvate and 5 mM malate (#P4562 and #M8304, respectively, Sigma-Aldrich, St. Louis, MO, USA) or 20 mM succinate (#S7501, Sigma-Aldrich, St. Louis, MO, USA) were added to stimulate the two different respiratory pathways. The reaction was observed, using the luciferin/luciferase chemiluminescent method (luciferin/luciferase ATP bioluminescence assay kit CLS II, Roche, Switzerland), with a luminometer (GloMax^®^ 20/20 Luminometer, Promega Italia, Italy), for 2 min every 30 s. For the calibration, ATP standard solutions in a concentration range between 10^−8^ and 10^−5^ M were used. Data were expressed as nmol ATP/min/10^6^ cells. 

### 4.5. P/O Ratio 

The P/O value was calculated as the ratio between the aerobic synthesized ATP and the consumed oxygen and represents a measure of OxPhos efficiency. Efficient mitochondria have a P/O value of around 2.5 or 1.5 when stimulated with pyruvate and malate or succinate, respectively, as respiring substrates. A P/O ratio lower than 2.5 for pyruvate and malate or lower than 1.5 for malate indicates that oxygen is not completely used for energy production but contributes to reactive oxygen species (ROS) production [45]. 

### 4.6. Electron Microscopy Analysis

FA and FAcorr lymphoblast and fibroblast pellets were fixed with 2.5% glutaraldehyde 0.1 M cacodylate buffer pH 7.6 for 1 h at room temperature. After post-fixation with 1% OsO4 in cacodylate buffer for 1 h, pellets were dehydrated in an ethanol series and embedded in Epon resin. Ultrathin sections stained with uranyl-acetate and lead citrate were observed with a Jeol Jem-1011 transmission electron microscope [14]. 

### 4.7. Cell Homogenate Preparation 

Fibroblast cell lines, which grow in adhesion, were detached from the culture flask by trypsinization for 5 min at 37 °C, after the culture medium removal and a wash in PBS (#14190250, ThermoFisher Scientifics, Waltham, MA, USA) to remove any traces of FBS. Then, trypsin (Trypsin-EDTA 1X in PBS, #ECB3052D, Euroclone, Milano, Italy) was blocked with a fresh culture medium, and the cells were collected. Lymphoblastoid cell lines growing in suspension were only collected. All cells were then centrifuged at 1000× *g* for 5 min to remove the growth medium. Next, pellets were washed twice in PBS and centrifuged again. All pellets were resuspended in an appropriate volume of Milli-Q water and sonicated (Microson XL Model DU-2000, Misonix Inc., Farmingdale, NY, USA) twice for 10 seconds each with a 30-second interval in between and in ice to prevent heating. Total protein content was evaluated according to the Bradford method [46]. 

### 4.8. Western Blot Analysis 

To perform denaturing electrophoresis (SDS-PAGE), 30 μg of proteins were loaded for each sample on 4–20% gradient gels (#4561094, BioRad, Hercules, CA, USA). The following primary antibodies were used: anti-ND1 (#ab181848, Abcam, Cambridge, UK), anti-SDHB (#ab84622, Abcam, UK), anti-MTCO2 (#ab79393, Abcam, Cambridge, UK), anti-ATP synthase β (#HPA001520, Sigma-Aldrich, St. Louis, MO, USA), anti-UCP2 (#sc-6525, Santa Cruz Biotechnology, Dallas, TX, USA), anti-DRP1 (#ab184247, Abcam, Cambridge, UK), anti-OPA1 (#HPA036926, Sigma-Aldrich, St. Louis, MO, USA), anti-MFN2 (#11925S, Cell Signaling Technology, USA), anti-Beclin1 (#3495P, Cell Signaling Technology, USA), anti-Atg7 (#8558P, Cell Signaling Technology, USA), anti-Atg12 (#4180P, Cell Signaling Technology, Danvers, MA, USA), anti-Atg16L1 (#8089P, Cell Signaling Technology, Danvers, MA, USA), anti-LC3 (#NB100-2220, Novus Biologicals, Centennial, CO, USA), anti-CLUH (#A301-764A Bethyl Lab. Inc., USA), anti-Pink1 (#PA1-4515, Invitrogen, USA), anti-Parkin (#PA5-13399 ThermoFisher, Waltham, MA, USA), and anti-Actin (#sc-1616, Santa Cruz Biotechnology, Dallas, TX, USA). All primary antibodies were diluted 1:1000 in PBS plus 0.15% Tween (PBSt, Tween was from Roche, Basilea, Switzerland, # 11332465001). Specific secondary antibodies were employed (#A0168 and # SAB3700870, Sigma-Aldrich, St. Louis, MO, USA), all diluted 1:10,000 in PBSt. Bands were detected and analyzed for optical density using an enhanced chemiluminescence substrate (ECL, # 1705061, BioRad, Hercules, CA, USA), a chemiluminescence system (Alliance 6.7 WL 20M, UVITEC., Cambridge, UK), and UV1D software (UVITEC, Cambridge, UK). All the bands of interest were normalized with Actin levels detected on the same membrane. 

### 4.9. Electron Transport between Complexes I and III Evaluation

The electron transfer between respiratory complex I and III was analyzed spectrophotometrically, following the reduction of cytochrome c, at 550 nm. For each assay, 50 μg of total protein was used. The reaction mix contained 100 mM Tris-HCl pH 7.4 and 0.03% of oxidized cytochrome c (#C2867, Sigma-Aldrich, St. Louis, MO, USA). The assay started with the addition of 0.7 mM NADH. If the electron transport between Complex I and Complex III is conserved the electrons pass from NADH to Complex I, then to Complex III via coenzyme Q, and finally to cytochrome c [12]. 

### 4.10. Lipid Content Evaluation

The lipid content was evaluated by the Sulfo-Phospho-Vanillin assay. Briefly, samples were incubated with 95% sulfuric acid at 95 °C for 20 min, quickly cooled, and evaluated at 535 nm. Afterward, a solution of 0.2 mg/mL vanillin (#V1104, Sigma-Aldrich, St. Louis, MO, USA) in 17% aqueous phosphoric acid was added to the samples, incubated for 10 min in the dark, and reevaluated at 535 nm. A mix of triglycerides (#17810, Sigma-Aldrich, St. Louis, MO, USA) was used to obtain a standard curve [12]. 

### 4.11. Malondialdehyde Level Evaluation

The malondialdehyde (MDA) concentration was evaluated by the thiobarbituric acid reactive substances (TBARS) assay. This test is based on the reaction of MDA, a breakdown product of lipid peroxides, with thiobarbituric acid (#T5500, Sigma-Aldrich, St. Louis, MO, USA). The TBARS solution contained 15% trichloroacetic acid in 0.25 N HCl and 26 mM thiobarbituric acid. To evaluate the basal concentration of MDA, 600 μL of TBARS solution was added to 50 μg of total protein dissolved in 300 μL of Milli-Q water. The mix was incubated for 40 min at 95 °C. After the sample was centrifuged at 20,000× *g* for 2 min, the supernatant was analyzed spectrophotometrically at 532 nm [12]. 

### 4.12. Confocal Microscopy Analysis

Cells were cultured in chamber-slides for 24 h in the absence or presence of P110. After PBS washes, cells were fixed with 0.3% paraformaldehyde (#P6148, Sigma-Aldrich, St. Louis, MO, USA) and permeabilized with 0.1% triton (#X100, Sigma-Aldrich, St. Louis, MO, USA). Cells were incubated overnight at 4 °C with the antibody against TOM20 (#42406S, Cell Signaling Technology, Danvers, MA, USA). After PBS washes, cells were incubated for 1 h at 25 °C with the Alexa-546-conjugated anti-rabbit antiserum (#A11010, Invitrogen, Waltham, MA, USA) as a secondary antibody. After PBS wash, chamber slides were mounted in mowiol. Immunofluorescence confocal laser scanner microscopy (CLSM) imaging was performed using a laser scanning spectral confocal microscope TCS SP2 AOBS (Leica, Wetzlar, Germany), equipped with Argon ion, He–Ne 543 nm, and He–Ne 633 nm lasers. Images were acquired through a HCX PL APO CS 40×/1.25 oil UV objective and processed with Leica. Images were acquired as single transcellular optical sections.

To evaluate the mitochondrial network shape, fibroblasts were scored depending on the morphology of most of their mitochondrial population as elongated or intermediated/short following the method described in [47].

### 4.13. Statistical Analysis

Data were analyzed appropriately using unpaired *t*-test or one-way ANOVA, using Prism 8 Software. Data are expressed as mean ± standard deviation (SD) and are representative of at least three independent experiments. An error with a probability of *p* < 0.05 was considered significant.

## Figures and Tables

**Figure 1 ijms-24-06557-f001:**
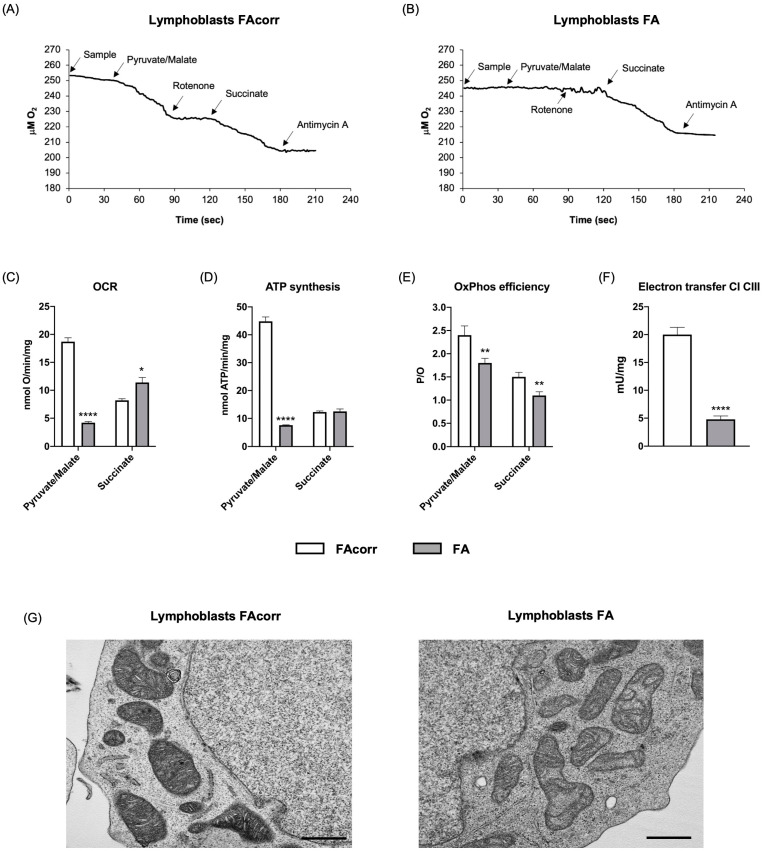
Aerobic metabolism and mitochondria structure in lymphoblasts mutated for the *FANC-A* gene. Oximetric traces of FAcorr (**A**) and FA (**B**) lymphoblasts. Arrows indicate the respiratory substrates or inhibitors’ addition. (**C**) Oxygen consumption rate (OCR). (**D**) Aerobic ATP synthesis. (**E**) P/O ratio, representing an OxPhos efficiency marker. (**F**) Electron transfer between respiratory complexes I and III. (**G**) Representative electron microscopy imaging of FAcorr and FA lymphoblasts to evaluate the mitochondrial morphology. Black scale bars correspond to 1 µm. Data reported in Panels A–E were obtained using pyruvate/malate or succinate as respiring substrates. Data reported in Panels A–F are reported as mean ± SD, and each panel is representative of at least three independent experiments. Statistical significance was tested opportunely with an unpaired *t*-test or one-way ANOVA; *, **, and **** represent a significant difference for *p* < 0.05, 0.01, or 0.0001, respectively, between FA and FAcorr cells.

**Figure 2 ijms-24-06557-f002:**
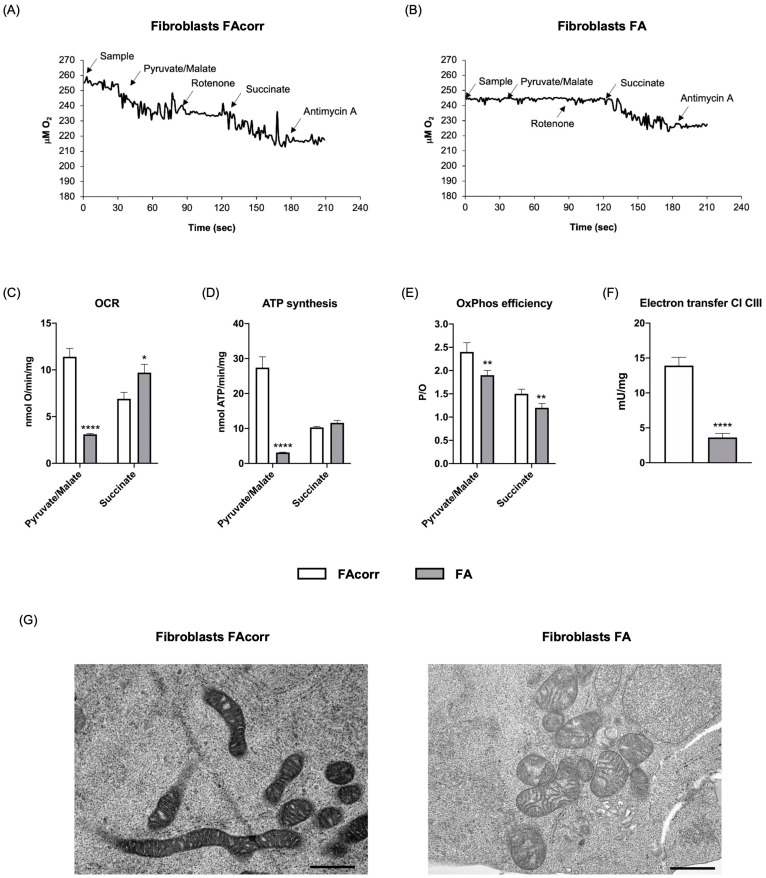
Aerobic metabolism in fibroblasts mutated for the *FANC-A* gene. Oximetric traces of FAcorr (**A**) and FA (**B**) fibroblasts. Arrows indicate the respiratory substrates or inhibitors’ addition. (**C**) Oxygen consumption rate (OCR). (**D**) Aerobic ATP synthesis. (**E**) P/O ratio, representing an OxPhos efficiency marker. (**F**) Electron transfer between respiratory complexes I and III. (**G**) Representative electron microscopy imaging of FAcorr and FA fibroblasts to evaluate the mitochondrial morphology. Black scale bars correspond to 1 µm. Data reported in Panels A-E were obtained using pyruvate/malate or succinate as respiring substrates. Data are reported as mean ± SD, and each panel is representative of at least three independent experiments. Statistical significance was tested opportunely with an unpaired *t*-test or one-way ANOVA; *, **, and **** represent a significant difference for *p* < 0.05, 0.01, or 0.0001, respectively, between FA and FAcorr cells.

**Figure 3 ijms-24-06557-f003:**
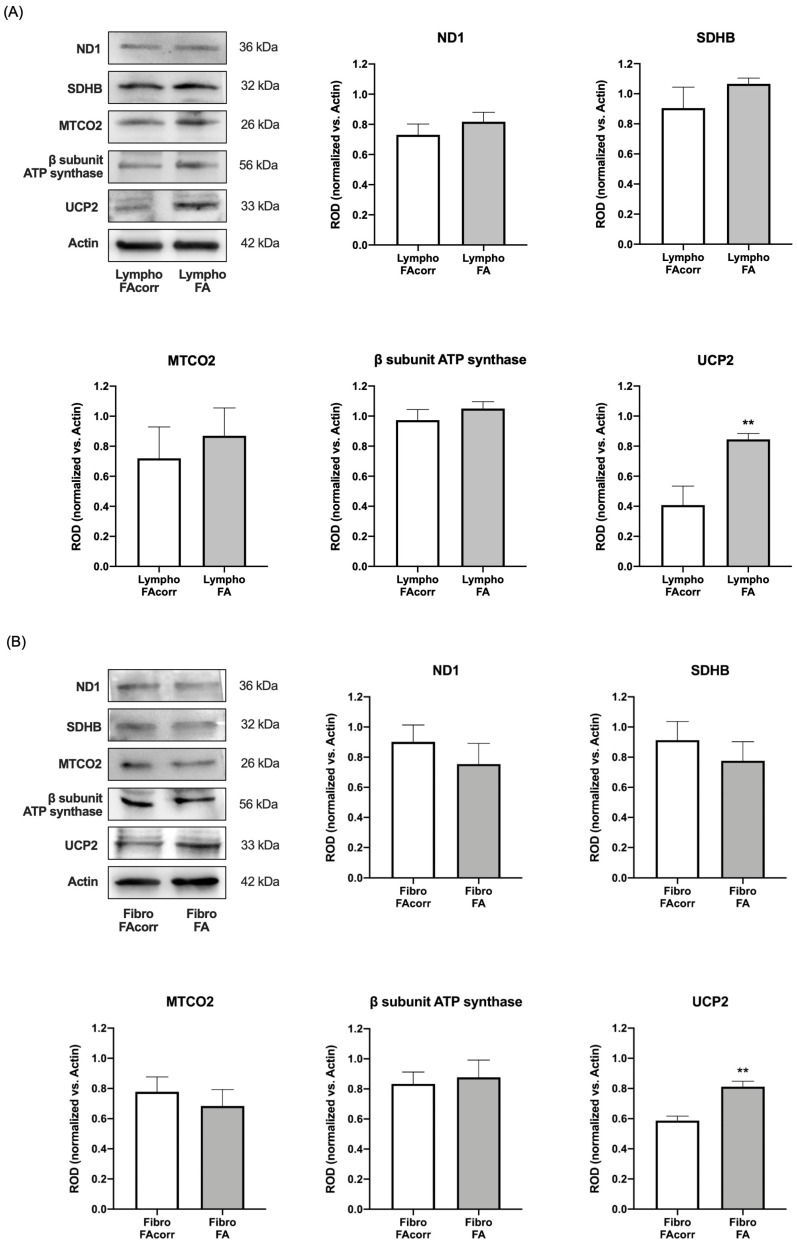
OxPhos subunits and UCP2 expression in lymphoblasts and fibroblasts mutated for the *FANC-A* gene. Western blot (WB) signals of OxPhos subunits expression in lymphoblast (**A**) and fibroblast (**B**) cell lines with and without *FANC-A* mutation. Protein expression levels are normalized on the housekeeping signal (Actin) revealed on the same membrane: ND1 (Complex I), SDHB (Complex II), MTCO2 (Complex III), and ATP synthase β subunit; WB signal of uncoupling protein 2 (UCP2) expression. Data in histograms are reported as mean ± SD, and each panel is representative of at least three independent experiments. Statistical significance was tested opportunely with an unpaired *t*-test; ** represents a significant difference for *p* < 0.01 between FA and FAcorr cells.

**Figure 4 ijms-24-06557-f004:**
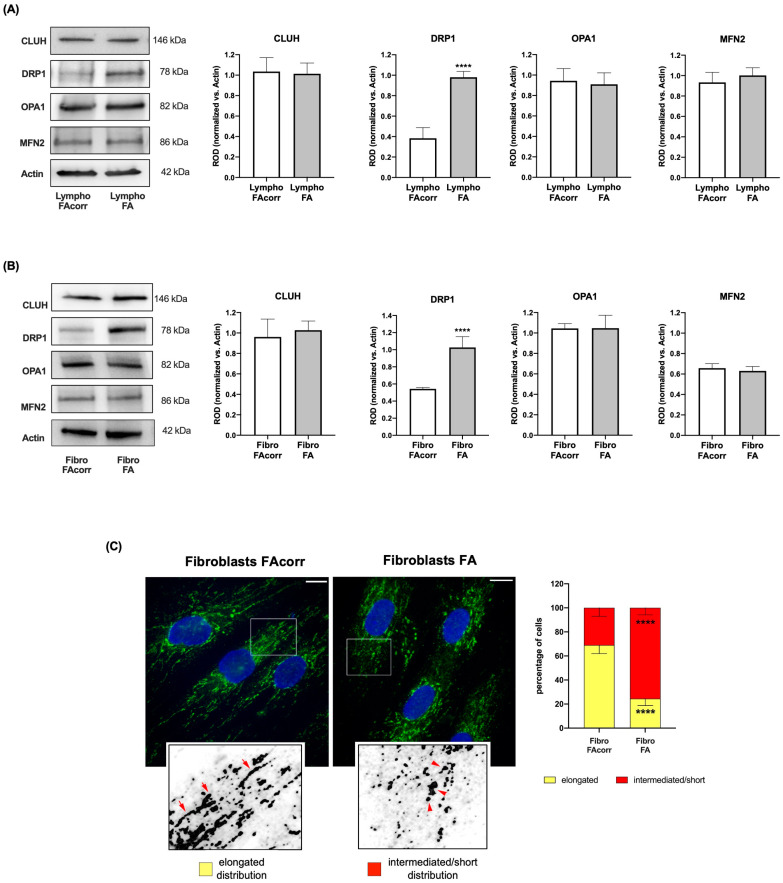
Expression levels of mitochondrial fusion and fission markers in lymphoblasts and fibroblasts mutated for the *FANC-A* gene. Lymphoblast (**A**) and fibroblast (**B**) cell lines expression of proteins involved in mitochondrial dynamics. WB signals of CLUH (mitochondrial biogenesis), DRP1 (mitochondrial fission), OPA1, MFN2 (mitochondrial fusion), and Actin (housekeeping protein used for signals normalization). For all the densitometry graphs, protein expression levels were normalized on the housekeeping signal, revealed on the same membrane. (**C**) Confocal imaging of FAcorr and FA fibroblasts stained with antibody against TOM20 (green) and DAPI (blue) to show the mitochondrial reticulum and nuclei, respectively. White scale bars correspond to 10 µm. The higher magnification insert corresponds to the area enclosed by the white square and is an example of mitochondrial network distribution. Fibroblasts were scored depending on the morphology of most of their mitochondrial population as elongated or intermediated/short. Results reported in the histogram on the right show that FA fibroblasts exhibited more often intermediated/short and less elongated mitochondria compared to FAcorr cells. Data are reported as mean ± SD. Histograms and WB signals are representative of at least three independent experiments. Statistical significance was tested with an unpaired *t*-test; **** represents a significant difference for *p* < 0.0001 between FA cells and FAcorr cells used as control.

**Figure 5 ijms-24-06557-f005:**
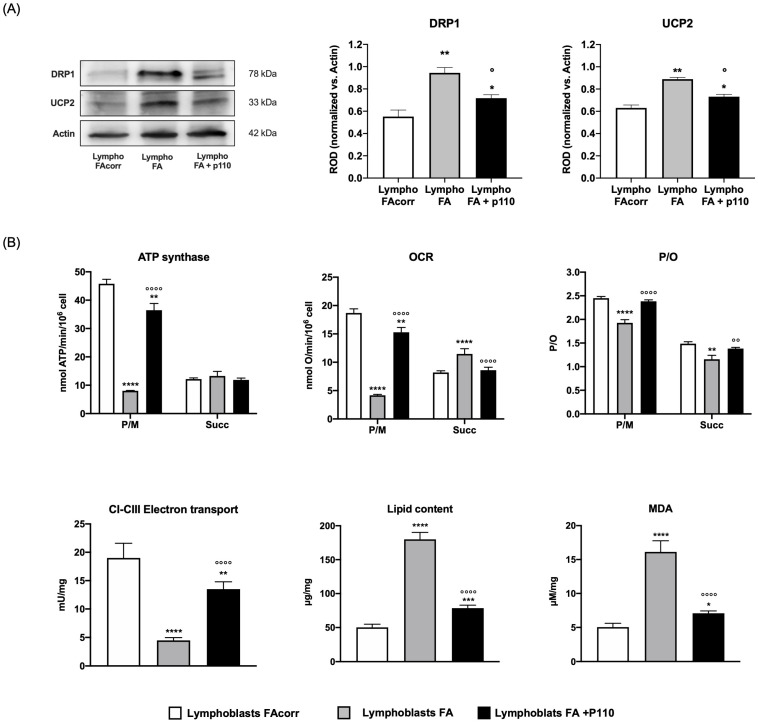
Effect of P110 treatment on DRP1 and UCP2 expression and energy metabolism in lymphoblasts mutated for the *FANC-A* gene. (**A**) WB signals of DRP1 (mitochondrial fission), UCP2 (uncoupling protein 2), and Actin (housekeeping protein used for signals normalization) expression in FAcorr lymphoblasts and FA lymphoblasts treated or not with P110. For all the densitometry graphs, protein expression levels were normalized on the housekeeping signal (Actin), revealed on the same membrane. (**B**) Energy and lipid metabolism in FAcorr and FA lymphoblasts treated or not with P110: ATP synthesis, oxygen consumption rate (OCR), and P/O value obtained after stimulation with pyruvate/malate (P/M) or succinate (Succ); electron transport between complexes I and III (CI-CIII); cellular lipid content; malondialdehyde (MDA) level, as a marker of lipid peroxidation. All the WB signals reported in (**A**) are representative of at least three independent experiments. In each panel, data are reported as mean ± SD, and each graph is representative of at least three independent experiments. Statistical significance was tested appropriately with a one-way ANOVA or an unpaired *t*-test; *, **, ***, and **** represent a significant difference for *p* < 0.05, 0.01, 0.001, and 0.0001 between FA cells and the FAcorr cells used as control; °, °°, and °°°° represent a significant difference for *p* < 0.05, 0.01, and 0.0001 between FA cells untreated and treated with P110.

**Figure 6 ijms-24-06557-f006:**
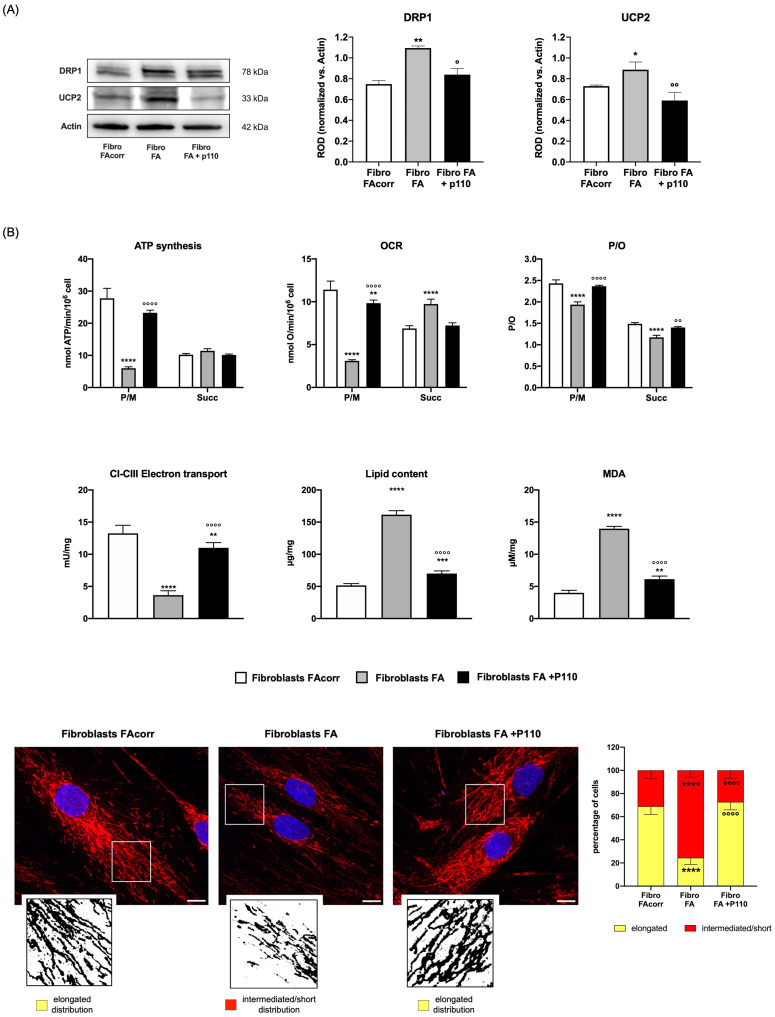
Effect of P110 treatment on DRP1 and UCP2 expression and energy metabolism in fibroblasts mutated for the *FANC-A* gene. (**A**) WB signals of DRP1 (mitochondrial fission), UCP2 (uncoupling protein 2), and Actin (housekeeping protein used for signals normalization) expression in FAcorr and FA fibroblasts treated or not with P110. For all the densitometry graphs protein expression levels were normalized on the housekeeping signal (Actin), revealed on the same membrane. (**B**) Energy and lipid metabolism in FAcorr and FA fibroblasts treated or not with P110: ATP synthesis, oxygen consumption rate (OCR), and P/O value obtained after stimulation with pyruvate/malate (P/M) or succinate (Succ); electron transport between complexes I and III (CI-CIII); cellular lipid content; malondialdehyde (MDA) level, as a marker of lipid peroxidation. (**C**) Confocal imaging of FAcorr fibroblasts and FA fibroblasts treated or not with P110 stained with antibody against TOM20 (red) and DAPI (blue), to show the mitochondrial reticulum and nuclei, respectively. White scale bars correspond to 10 µm. The higher magnification insert corresponds to the area enclosed by the white square and is an example of mitochondrial network distribution. Fibroblasts were scored depending on the morphology of most of their mitochondrial population as elongated or intermediated/short. Results reported in the histogram on the right show that P110 treatment on FA fibroblasts restores the percentage of cells with elongated distribution, reaching values similar to those of FAcorr cells. Data and WB signals are representative of at least three independent experiments. In each panel, data are reported as mean ± SD. Statistical significance was tested appropriately with a one-way ANOVA or an unpaired *t*-test; *, **, ***, and **** represent a significant difference for *p* < 0.05, 0.01, 0.001, and 0.0001 between FA cells and the FAcorr cells used as control; °, °°, and °°°° represent a significant difference for *p* < 0.05, 0.01, and 0.0001 between FA cells untreated and treated with P110.

**Figure 7 ijms-24-06557-f007:**
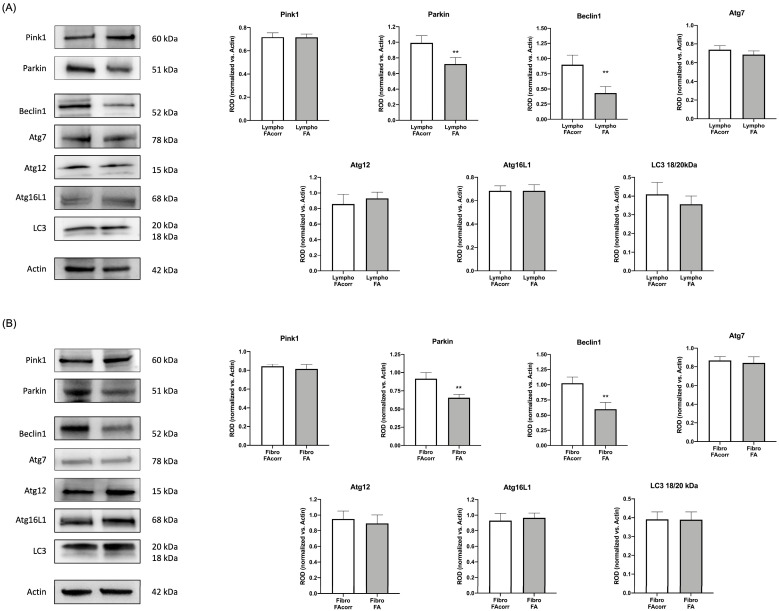
Expression levels of mitophagy and autophagy markers in lymphoblasts and fibroblasts mutated for the *FANC-A* gene. Lymphoblast (**A**) and fibroblast (**B**) cell lines expression of proteins involved in mitophagy and autophagy processes. WB signals of Pink1, Parkin, Beclin1, Atg7, Atg12, Atg16L1, and LC3. Actin signal has been used as a housekeeping signal for data normalization. The LC3 graph represents the ratio between the active (18 kDa) and inactive (20 kDa) form of the protein. Data are reported as mean ± SD. Histograms and WB signals are representative of at least three independent experiments. Statistical significance was tested with an unpaired *t*-test; ** represents a significant difference for *p* < 0.01 between FA cells and FAcorr cells used as control.

## Data Availability

The analyzed data supporting the conclusions of this article are included within this article and its additional files.

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
