# Peer review of "Altered Mitochondrial Dynamic in Lymphoblasts and Fibroblasts Mutated for FANCA-A Gene: The Central Role of DRP1"

_ijms, 2023, doi:10.3390/ijms24076557_

Round 1
Reviewer 1 Report
The manuscript focuses on Fanconi anemia disease and mitochondrial functions in case of pathology. Three fibroblast cell lines and three lymphoblast cell lines from FANC-A patients were used for the study. As a result, it is suggested that the changes in mitochondrial functions underly pathology of Fanconi anemia. The manuscript would benefit from a more clear conclusion of the study.
Specific points to consider:
1) Figure 1, panels B and C. For western blot, kindly indicate molecular weights determined using corresponding markers (kD).
2) Figure 2, panels B and C. For western blot, kindly indicate molecular weights determined using corresponding markers (kD).
3) Figure 3, panels B and C. For western blot, kindly indicate molecular weights determined using corresponding markers (kD).
4) Figure 4, panels B and C. For western blot, kindly indicate molecular weights determined using corresponding markers (kD).
5) Line 169. "," should be ".".
6) Figure 5, panel A. For western blot, kindly indicate molecular weights determined using corresponding markers (kD).
7) Figure 6, panel A. For western blot, kindly indicate molecular weights determined using corresponding markers (kD).
8) In Materials and Methods, kindly indicate catalog numbers and manufacturing sites of all reagents used (it is made for some, but not for all).
9) Line 341. For cell collection using a centrifuge, it is not sufficient to specify 1500 rpm without indicating centrifuge and rotor type. Units "g" can be used instead, or together with a more detailed description of the centrifuge type as well as the type of rotor and rpm.
10) Line 338. For "trypsinization" kindly indicate the concentration of trypsin and its origin.
11) Line 384. "14000 rpm" is not sufficient. Kindly indicate centrifuge and rotor type and/or express it in "g".
12) Different ways to write thousands is used, e.g. 10,000 (dilution for secondary antibodies) and 14000 (rpm). Kindly be consistent and choose one system.
Optional:
13) The ending of the introduction, i.e. lines 58-62, is traditionally used to summarize the findings and conclusions of the study, not the aims. Although it is up to the authors whether to keep the last paragraph of the Introduction the way it is now or to modify it.
Author Response
The manuscript focuses on Fanconi anemia disease and mitochondrial functions in case of pathology. Three fibroblast cell lines and three lymphoblast cell lines from FANC-A patients were used for the study. As a result, it is suggested that the changes in mitochondrial functions underly pathology of Fanconi anemia. The manuscript would benefit from a more clear conclusion of the study.
Specific points to consider:
Q1) Figure 1, panels B and C. For western blot, kindly indicate molecular weights determined using corresponding markers (kD).
R1) As requested, the molecular weight of the band shown in the western blot analysis was indicated in the revised version.
Q2) Figure 2, panels B and C. For western blot, kindly indicate molecular weights determined using corresponding markers (kD).
R2) As requested, the molecular weight of the band shown in the western blot analysis was indicated in the revised version.
Q3) Figure 3, panels B and C. For western blot, kindly indicate molecular weights determined using corresponding markers (kD).
R3) As requested, the molecular weight of the band shown in the western blot analysis was indicated in the revised version.
Q4) Figure 4, panels B and C. For western blot, kindly indicate molecular weights determined using corresponding markers (kD).
R4) As requested, the molecular weight of the band shown in the western blot analysis was indicated in the revised version.
Q5) Line 169. "," should be ".".
R5) The typo has been corrected
Q6) Figure 5, panel A. For western blot, kindly indicate molecular weights determined using corresponding markers (kD).
R6) As requested, the molecular weight of the band shown in the western blot analysis was indicated in the revised version.
Q7) Figure 6, panel A. For western blot, kindly indicate molecular weights determined using corresponding markers (kD).
R7) As requested, the molecular weight of the band shown in the western blot analysis was indicated in the revised version.
Q8) In Materials and Methods, kindly indicate catalog numbers and manufacturing sites of all reagents used (it is made for some, but not for all).
R8) We apologize for the lack. In the revised version, catalog numbers and manufacturing sites were added for all reagents.
Q9) Line 341. For cell collection using a centrifuge, it is not sufficient to specify 1500 rpm without indicating centrifuge and rotor type. Units "g" can be used instead, or together with a more detailed description of the centrifuge type as well as the type of rotor and rpm.
R9) We apologize for the mistake. In the revised version, “rpm” value has been substituted with “g” value.
Q10) Line 338. For "trypsinization" kindly indicate the concentration of trypsin and its origin.
R10) We apologize for the lack. In the revised version, we have specified the trypsin catalog number and the used concentration.
Q11) Line 384. "14000 rpm" is not sufficient. Kindly indicate centrifuge and rotor type and/or express it in "g".
R11) We apologize for the mistake. In the revised version, “rpm” value has been substituted with “g” value.
Q12) Different ways to write thousands is used, e.g. 10,000 (dilution for secondary antibodies) and 14000 (rpm). Kindly be consistent and choose one system.
R12) We apologize for the mistake. In the revised version, numbers have been indicated adding a comma to separate the thousands
Optional:
Q13) The ending of the introduction, i.e. lines 58-62, is traditionally used to summarize the findings and conclusions of the study, not the aims. Although it is up to the authors whether to keep the last paragraph of the Introduction the way it is now or to modify it.
R13) We thank the Reviewer for the advice. In the revised version, we have maintained the description of the manuscript's aims and added a summary of the obtained results.

Reviewer 2 Report
In this study entitled "Altered mitochondrial dynamic and quality control in lymphoblasts and fibroblast mutated for FANCA-A gene: the central 3 role of DRP1.", Bertola et al aimed to deeply investigate whether the mitochondrial metabolism alterations observed in FA depend on altered expression of proteins belonging to the oxidative phosphorylation (OxPhos) machinery or modulators of mitochondrial biogenesis, dynamics, and quality control in lymphoblasts and fibroblasts carrying the mutated FANC-A gene, comparing the results with isogenic corrected FANC-A gene cell lines.
This is a highly descriptive study and conclusions are not supported by data.
Main point are the following:
1. It is not clear to the reviewer how data reported in figure 1A have been obtained. Representative curve for OCR should be added.
2. Figures 2A, 3A and 4A are unuseful and should be removed.
3. The authors claim that the dysfunctional OxPhos is associated with oxidative stress enhancement, which 99
creates a vicious circle that leads to damage to the mitochondria (27,28). However, they did not show whether in their models mitochondrial alterations are present. Mitochondrial ultrastructure should be visualized by transmission electron microscopy.
4. Again, the authors claim that FANC-A cells display unfunctional mitophagy and autophagy, but they concluded this by merely analysing the expression levels of some proteins involved in mitothagy and autophagy. In this manuscript there are not experiments reporting the presence of mitophagy in any cell model.
Concerning autophagy, according to the “Guidelines for the use and interpretation of assays for monitoring autophagy (4th edition)” by Klionsky et al, neither assessment of total LC3, LC3-I consumption nor the evaluation of LC3-II levels would necessarily reveal a slight induction of autophagy. Thus I would recommend to measure the autophagic flux in both the presence and absence of lysosomal, or vacuolar degradation. Lysosomal degradation can be prevented through the use of protease inhibitors (e.g., pepstatin A, leupeptin and E-64d), compounds that neutralize the lysosomal pH such as bafilomycin A1, chloroquine or NH4Cl, or by treatment with agents that block the fusion of autophagosomes with lysosomes. Alternatively, knocking down or knocking out LAMP2 (lysosomal associated membrane protein 2) represents a genetic approach to block the fusion of autophagosomes and lysosomes.
Autophagy should be also monitored by transmission electron microscopy as it allows the visualization of the sequential morphological changes during the autophagic process. Also in this case it is possible it is possible to inhibit the fusion of autophago- somes and lysosomes using bafilomycin A1.
5. In figure 4D representative confocal microscopy images of mitochondrial network are reported. However quantification is missing. This should be added to conclude that a disruption of mitochondrial network is present. Same for figure 6C.
Author Response
In this study entitled "Altered mitochondrial dynamic and quality control in lymphoblasts and fibroblast mutated for FANCA-A gene: the central 3 role of DRP1.", Bertola et al aimed to deeply investigate whether the mitochondrial metabolism alterations observed in FA depend on altered expression of proteins belonging to the oxidative phosphorylation (OxPhos) machinery or modulators of mitochondrial biogenesis, dynamics, and quality control in lymphoblasts and fibroblasts carrying the mutated FANC-A gene, comparing the results with isogenic corrected FANC-A gene cell lines.
This is a highly descriptive study and conclusions are not supported by data.
Main point are the following:
Q1. It is not clear to the reviewer how data reported in figure 1A have been obtained. Representative curve for OCR should be added.
R1 We apologize for the unclear description of how we obtained the data shown in Figure 1A. The table reports the oxygen consumption rate (OCR), the aerobic ATP synthesis, and the consequent P/O ratio as markers of mitochondrial aerobic metabolism quality. As reported in the Material and Method section, an oxygen microsensor and a luminometer were employed for this scope. Pyruvate plus malate and succinate have been used as respiratory substrates to distinguish the functionality of the pathway triggered by Complex I and Complex II, respectively.
Indeed, we have already published data on mitochondrial function in Fanconi cells several times, showing, in some cases, the oximetry traces [1–9]. Specifically, our previous data show that the metabolic defect consists of altered electron transport between Complexes I and III [1,2,5]
Therefore, in this manuscript, the metabolic data have only been included in a table as a starting point for investigating the role of mitochondrial dynamics and quality control in FA.
However, in the revised submission, we have included an example of oximetry traces of FAcorr and FA lymphoblasts as material for the Reviewer.
Q2. Figures 2A, 3A and 4A are unuseful and should be removed.
R2. We regret that the Reviewer considers the diagrams in Figures 2, 3, and 4 to be minute. Since the pathways we have described are composed of many different proteins, we believe it would be helpful to focus on where the proteins analyzed with WB are involved. Therefore, we prefer to maintain the patterns shown in Figures 2, 3, and 4.
Q3. The authors claim that the dysfunctional OxPhos is associated with oxidative stress enhancement, which creates a vicious circle that leads to damage to the mitochondria (27,28). However, they did not show whether in their models mitochondrial alterations are present. Mitochondrial ultrastructure should be visualized by transmission electron microscopy.
R3. Indeed, several previous papers already report the altered mitochondrial structure in the same FA cellular models used for this manuscript [2,3,10–12], as mentioned in the introduction section. For this reason, we prefer not to insert in this manuscript a data repetition.
Q4. Again, the authors claim that FANC-A cells display unfunctional mitophagy and autophagy, but they concluded this by merely analysing the expression levels of some proteins involved in mitothagy and autophagy. In this manuscript there are not experiments reporting the presence of mitophagy in any cell model.
Concerning autophagy, according to the “Guidelines for the use and interpretation of assays for monitoring autophagy (4th edition)” by Klionsky et al, neither assessment of total LC3, LC3-I consumption nor the evaluation of LC3-II levels would necessarily reveal a slight induction of autophagy. Thus I would recommend to measure the autophagic flux in both the presence and absence of lysosomal, or vacuolar degradation. Lysosomal degradation can be prevented through the use of protease inhibitors (e.g., pepstatin A, leupeptin and E-64d), compounds that neutralize the lysosomal pH such as bafilomycin A1, chloroquine or NH4Cl, or by treatment with agents that block the fusion of autophagosomes with lysosomes. Alternatively, knocking down or knocking out LAMP2 (lysosomal associated membrane protein 2) represents a genetic approach to block the fusion of autophagosomes and lysosomes.
Autophagy should be also monitored by transmission electron microscopy as it allows the visualization of the sequential morphological changes during the autophagic process. Also in this case it is possible it is possible to inhibit the fusion of autophago- somes and lysosomes using bafilomycin A1.
R4. We agree with Reviewer that our data do not show cell biology data on autophagy and mitophagy functionality. However, this was not the purpose of the manuscript since the literature already reports that mitophagy and autophagy are defective in FA cells [13–16]. Conversely, we evaluated the expression of some key proteins in the processes of mitophagy and autophagy to understand whether the defect was in the pathway's triggers or in its effectors. In the revised version, we have better explained our purpose.
Q5. In figure 4D representative confocal microscopy images of mitochondrial network are reported. However quantification is missing. This should be added to conclude that a disruption of mitochondrial network is present. Same for figure 6C.
R5. We thank the Reviewer for this suggestion. In the revised version, to quantify the different organization of the mitochondrial reticulum in P110-treated FAcorr, FA, and FA fibroblasts, we added a histogram to Figures 4 and 6 showing the percentage of cells with elongated or intermediate/short mitochondrial structures following the method reported by Lopez-Domenech G, et al. [17].
References for Reviewer:
- Cappelli, E.; Cuccarolo, P.; Stroppiana, G.; Miano, M.; Bottega, R.; Cossu, V.; Degan, P.; Ravera, S. Defects in Mitochondrial Energetic Function Compels Fanconi Anaemia Cells to Glycolytic Metabolism. Biochim Biophys Acta Mol Basis Dis2017, 1863, doi:10.1016/j.bbadis.2017.03.008.
- Ravera, S.; Vaccaro, D.; Cuccarolo, P.; Columbaro, M.; Capanni, C.; Bartolucci, M.; Panfoli, I.; Morelli, A.; Dufour, C.; Cappelli, E.; et al. Mitochondrial Respiratory Chain Complex i Defects in Fanconi Anemia Complementation Group A. Biochimie2013, 95, doi:10.1016/j.biochi.2013.06.006.
- Ravera, S.; Cossu, V.; Tappino, B.; Nicchia, E.; Dufour, C.; Cavani, S.; Sciutto, A.; Bolognesi, C.; Columbaro, M.; Degan, P.; et al. Concentration-Dependent Metabolic Effects of Metformin in Healthy and Fanconi Anemia Lymphoblast Cells. J Cell Physiol2018, 233, doi:10.1002/jcp.26085.
- Bottega, R.; Nicchia, E.; Cappelli, E.; Ravera, S.; de Rocco, D.; Faleschini, M.; Corsolini, F.; Pierri, F.; Calvillo, M.; Russo, G.; et al. Hypomorphic FANCA Mutations Correlate with Mild Mitochondrial and Clinical Phenotype in Fanconi Anemia. Haematologica2018, 103, doi:10.3324/haematol.2017.176131.
- Cappelli, E.; Degan, P.; Bruno, S.; Pierri, F.; Miano, M.; Raggi, F.; Farruggia, P.; Mecucci, C.; Crescenzi, B.; Naim, V.; et al. The Passage from Bone Marrow Niche to Bloodstream Triggers the Metabolic Impairment in Fanconi Anemia Mononuclear Cells. Redox Biol2020, 36, doi:10.1016/j.redox.2020.101618.
- Cappelli, E.; Bertola, N.; Bruno, S.; Degan, P.; Regis, S.; Corsolini, F.; Banelli, B.; Dufour, C.; Ravera, S. A Multidrug Approach to Modulate the Mitochondrial Metabolism Impairment and Relative Oxidative Stress in Fanconi Anemia Complementation Group A. Metabolites2021, 12, 6, doi:10.3390/metabo12010006.
- Lyakhovich, A. Damaged Mitochondria and Overproduction of ROS in Fanconi Anemia Cells. Rare Diseases2013, 1, e24048–e24048, doi:10.4161/rdis.24048.
- Kumari, U.; Ya Jun, W.; Huat Bay, B.; Lyakhovich, A. Evidence of Mitochondrial Dysfunction and Impaired ROS Detoxifying Machinery in Fanconi Anemia Cells. Oncogene2014, 33, 165–172, doi:10.1038/onc.2012.583.
- Pagano, G.; Talamanca, A.A.; Castello, G.; d’Ischia, M.; Pallardó, F. v; Petrović, S.; Porto, B.; Tiano, L.; Zatterale, A. Bone Marrow Cell Transcripts from Fanconi Anaemia Patients Reveal in Vivo Alterations in Mitochondrial, Redox and DNA Repair Pathways. Eur J Haematol2013, 91, 141–151, doi:10.1111/ejh.12131.
- Capanni, C.; Bruschi, M.; Columbaro, M.; Cuccarolo, P.; Ravera, S.; Dufour, C.; Candiano, G.; Petretto, A.; Degan, P.; Cappelli, E. Changes in Vimentin, Lamin A/C and Mitofilin Induce Aberrant Cell Organization in Fibroblasts from Fanconi Anemia Complementation Group A (FA-A) Patients. Biochimie2013, 95, doi:10.1016/j.biochi.2013.06.024.
- Ravera, S.; Degan, P.; Sabatini, F.; Columbaro, M.; Dufour, C.; Cappelli, E. Altered Lipid Metabolism Could Drive the Bone Marrow Failure in Fanconi Anaemia. Br J Haematol2019, 184, doi:10.1111/bjh.15171.
- Columbaro, M.; Ravera, S.; Capanni, C.; Panfoli, I.; Cuccarolo, P.; Stroppiana, G.; Degan, P.; Cappell, E. Treatment of FANCA Cells with Resveratrol and N-Acetylcysteine: A Comparative Study. PLoS One2014, 9, doi:10.1371/JOURNAL.PONE.0104857.
- Sumpter, R.; Levine, B. Emerging Functions of the Fanconi Anemia Pathway at a Glance. J Cell Sci2017, 130, 2657–2662, doi:10.1242/jcs.204909.
- Shyamsunder, P.; Esner, M.; Barvalia, M.; Wu, Y.J.; Loja, T.; Boon, H.B.; Lleonart, M.E.; Verma, R.S.; Krejci, L.; Lyakhovich, A. Impaired Mitophagy in Fanconi Anemia Is Dependent on Mitochondrial Fission. Oncotarget2016, 7, 58065, doi:10.18632/ONCOTARGET.11161.
- Sumpter, R.; Sirasanagandla, S.; Fernández, Á.F.; Wei, Y.; Dong, X.; Franco, L.; Zou, Z.; Marchal, C.; Lee, M.Y.; Clapp, D.W.; et al. Fanconi Anemia Proteins Function in Mitophagy and Immunity. Cell2016, 165, 867–881, doi:10.1016/j.cell.2016.04.006.
- Sumpter, R.; Levine, B. Novel Functions of Fanconi Anemia Proteins in Selective Autophagy and Inflammation. Oncotarget2016, 7, 50820–50821, doi:10.18632/ONCOTARGET.10970.
- López‐Doménech, G.; Covill‐Cooke, C.; Ivankovic, D.; Halff, E.F.; Sheehan, D.F.; Norkett, R.; Birsa, N.; Kittler, J.T. Miro Proteins Coordinate Microtubule- and Actin-Dependent Mitochondrial Transport and Distribution. EMBO J2018, 37, 321–336, doi:10.15252/EMBJ.201696380.

Round 2
Reviewer 2 Report
The authors did not address most of my concerns sufficiently to make this manuscript suitable for publication.
Author Response
In this study entitled "Altered mitochondrial dynamic and quality control in lymphoblasts and fibroblast mutated for FANCA-A gene: the central 3 role of DRP1.", Bertola et al aimed to deeply investigate whether the mitochondrial metabolism alterations observed in FA depend on altered expression of proteins belonging to the oxidative phosphorylation (OxPhos) machinery or modulators of mitochondrial biogenesis, dynamics, and quality control in lymphoblasts and fibroblasts carrying the mutated FANC-A gene, comparing the results with isogenic corrected FANC-A gene cell lines.
This is a highly descriptive study and conclusions are not supported by data.
Main point are the following:
Q1. It is not clear to the reviewer how data reported in figure 1A have been obtained. Representative curve for OCR should be added.
R1 To better clarify the data reported in Figure 1A, in the revised version, we have added two new figures reporting both for lymphoblasts and fibroblasts: (i) representative OCR curves for FAcorr and FA recorded by an oxygen microsensor; (ii) the amount of consumed oxygen normalized on 10^6 cells; (iii) the amount of synthesized ATP normalized on 10^6 cells obtained through luminometric analyses; (iv) P/O values calculated as the ratio between the consumed oxygen and synthesized ATP, indicating the OxPhos efficiency; (v) the electron transport between complex I and complex III. In addition, we have added, in revised Figure 1, a representative electron microscopy image of FAcorr e FA lymphoblasts, in which is evident the altered mitochondria structure in the cells carrying the FANC-A gene mutation.All the biochemical items confirm the altered mitochondrial metabolism typical of FA cells, confirming the previously published data [1–9].
Q2. Figures 2A, 3A and 4A are unuseful and should be removed.
R2. As requested by the Reviewer, the schemes corresponding to the 2A, 3A, and 4A panels of the original paper version have been eliminated.
Q3. The authors claim that the dysfunctional OxPhos is associated with oxidative stress enhancement, which creates a vicious circle that leads to damage to the mitochondria (27,28). However, they did not show whether in their models mitochondrial alterations are present. Mitochondrial ultrastructure should be visualized by transmission electron microscopy.
R3. As requested by the Reviewer, we have added, in the new Figures 1 and 2, a representative electron microscopy image of mitochondria in FAcorr e FA lymphoblasts and fibroblasts, respectively, confirming the presence of damaged mitochondria in FA cells as already reported in [2,3,10–12].
Q4. Again, the authors claim that FANC-A cells display unfunctional mitophagy and autophagy, but they concluded this by merely analysing the expression levels of some proteins involved in mitothagy and autophagy. In this manuscript there are not experiments reporting the presence of mitophagy in any cell model.
Concerning autophagy, according to the “Guidelines for the use and interpretation of assays for monitoring autophagy (4th edition)” by Klionsky et al, neither assessment of total LC3, LC3-I consumption nor the evaluation of LC3-II levels would necessarily reveal a slight induction of autophagy. Thus I would recommend to measure the autophagic flux in both the presence and absence of lysosomal, or vacuolar degradation. Lysosomal degradation can be prevented through the use of protease inhibitors (e.g., pepstatin A, leupeptin and E-64d), compounds that neutralize the lysosomal pH such as bafilomycin A1, chloroquine or NH4Cl, or by treatment with agents that block the fusion of autophagosomes with lysosomes. Alternatively, knocking down or knocking out LAMP2 (lysosomal associated membrane protein 2) represents a genetic approach to block the fusion of autophagosomes and lysosomes.
Autophagy should be also monitored by transmission electron microscopy as it allows the visualization of the sequential morphological changes during the autophagic process. Also in this case it is possible it is possible to inhibit the fusion of autophago- somes and lysosomes using bafilomycin A1.
R4. Demonstrating the unfunctional mitophagy and autophagy in FA cells is not the purpose of this manuscript, as the literature already reports this claim [13–16]. Our aim consists of the expression evaluation of key proteins involved in these two processes by western blot analyses since the literature still does not identify the molecular cause(s) for mitophagy and autophagy alteration. However, to meet the reviewer's point of view, in the revised version, we have limited ourselves to reporting the finding of altered expression of Parkin and Beclin1 without insisting on the biological significance of this data.
Q5. In figure 4D representative confocal microscopy images of mitochondrial network are reported. However quantification is missing. This should be added to conclude that a disruption of mitochondrial network is present. Same for figure 6C.
R5. We thank the Reviewer for this suggestion. In the revised version, to quantify the different organization of the mitochondrial reticulum in P110-treated FAcorr, FA, and FA fibroblasts, we added a histogram to Figures 4 and 5 (in the revised version) showing the percentage of cells with elongated or intermediate/short mitochondrial structures following the method reported by Lopez-Domenech G, et al. [17].
References for Reviewer:
- Cappelli, E.; Cuccarolo, P.; Stroppiana, G.; Miano, M.; Bottega, R.; Cossu, V.; Degan, P.; Ravera, S. Defects in Mitochondrial Energetic Function Compels Fanconi Anaemia Cells to Glycolytic Metabolism. BiochimBiophys Acta Mol Basis Dis2017, 1863, doi:10.1016/j.bbadis.2017.03.008.
- Ravera, S.; Vaccaro, D.; Cuccarolo, P.; Columbaro, M.; Capanni, C.; Bartolucci, M.; Panfoli, I.; Morelli, A.; Dufour, C.; Cappelli, E.; et al. MitochondrialRespiratory Chain Complex i Defects in Fanconi Anemia Complementation Group A. Biochimie2013, 95, doi:10.1016/j.biochi.2013.06.006.
- Ravera, S.; Cossu, V.; Tappino, B.; Nicchia, E.; Dufour, C.; Cavani, S.; Sciutto, A.; Bolognesi, C.; Columbaro, M.; Degan, P.; et al. Concentration-Dependent Metabolic Effects of Metformin in Healthy and Fanconi Anemia Lymphoblast Cells. J Cell Physiol2018, 233, doi:10.1002/jcp.26085.
- Bottega, R.; Nicchia, E.; Cappelli, E.; Ravera, S.; de Rocco, D.; Faleschini, M.; Corsolini, F.; Pierri, F.; Calvillo, M.; Russo, G.; et al. Hypomorphic FANCA Mutations Correlate with Mild Mitochondrial and Clinical Phenotype in Fanconi Anemia. Haematologica2018, 103, doi:10.3324/haematol.2017.176131.
- Cappelli, E.; Degan, P.; Bruno, S.; Pierri, F.; Miano, M.; Raggi, F.; Farruggia, P.; Mecucci, C.; Crescenzi, B.; Naim, V.; et al. The Passage from Bone Marrow Niche to Bloodstream Triggers the Metabolic Impairment in Fanconi Anemia Mononuclear Cells. Redox Biol2020, 36, doi:10.1016/j.redox.2020.101618.
- Cappelli, E.; Bertola, N.; Bruno, S.; Degan, P.; Regis, S.; Corsolini, F.; Banelli, B.; Dufour, C.; Ravera, S. A Multidrug Approach to Modulate the Mitochondrial Metabolism Impairment and Relative Oxidative Stress in Fanconi Anemia Complementation Group A. Metabolites2021, 12, 6, doi:10.3390/metabo12010006.
- Lyakhovich, A. Damaged Mitochondria and Overproduction of ROS in Fanconi Anemia Cells. Rare Diseases2013, 1, e24048–e24048, doi:10.4161/rdis.24048.
- Kumari, U.; Ya Jun, W.; Huat Bay, B.; Lyakhovich, A. Evidence of Mitochondrial Dysfunction and Impaired ROS Detoxifying Machinery in Fanconi Anemia Cells. Oncogene2014, 33, 165–172, doi:10.1038/onc.2012.583.
- Pagano, G.; Talamanca, A.A.; Castello, G.; d’Ischia, M.; Pallardó, F. v; Petrović, S.; Porto, B.; Tiano, L.; Zatterale, A. Bone Marrow Cell Transcripts from Fanconi AnaemiaPatientsReveal in Vivo Alterations in Mitochondrial, Redox and DNA Repair Pathways. Eur J Haematol2013, 91, 141–151, doi:10.1111/ejh.12131.
- Capanni, C.; Bruschi, M.; Columbaro, M.; Cuccarolo, P.; Ravera, S.; Dufour, C.; Candiano, G.; Petretto, A.; Degan, P.; Cappelli, E. Changes in Vimentin, Lamin A/C and Mitofilin Induce Aberrant Cell Organization in Fibroblasts from Fanconi Anemia Complementation Group A (FA-A) Patients. Biochimie2013, 95, doi:10.1016/j.biochi.2013.06.024.
- Ravera, S.; Degan, P.; Sabatini, F.; Columbaro, M.; Dufour, C.; Cappelli, E. AlteredLipidMetabolismCould Drive the Bone MarrowFailure in Fanconi Anaemia. Br J Haematol2019, 184, doi:10.1111/bjh.15171.
- Columbaro, M.; Ravera, S.; Capanni, C.; Panfoli, I.; Cuccarolo, P.; Stroppiana, G.; Degan, P.; Cappell, E. Treatment of FANCA Cells with Resveratrol and N-Acetylcysteine: A Comparative Study. PLoS One2014, 9, doi:10.1371/JOURNAL.PONE.0104857.
- Sumpter, R.; Levine, B. Emerging Functions of the Fanconi Anemia Pathway at a Glance. J Cell Sci2017, 130, 2657–2662, doi:10.1242/jcs.204909.
- Shyamsunder, P.; Esner, M.; Barvalia, M.; Wu, Y.J.; Loja, T.; Boon, H.B.; Lleonart, M.E.; Verma, R.S.; Krejci, L.; Lyakhovich, A. Impaired Mitophagy in Fanconi Anemia Is Dependent on Mitochondrial Fission. Oncotarget2016, 7, 58065, doi:10.18632/ONCOTARGET.11161.
- Sumpter, R.; Sirasanagandla, S.; Fernández, Á.F.; Wei, Y.; Dong, X.; Franco, L.; Zou, Z.; Marchal, C.; Lee, M.Y.; Clapp, D.W.; et al. Fanconi Anemia Proteins Function in Mitophagy and Immunity. Cell2016, 165, 867–881, doi:10.1016/j.cell.2016.04.006.
- Sumpter, R.; Levine, B. Novel Functions of Fanconi Anemia Proteins in Selective Autophagy and Inflammation. Oncotarget2016, 7, 50820–50821, doi:10.18632/ONCOTARGET.10970.
- López‐Doménech, G.; Covill‐Cooke, C.; Ivankovic, D.; Halff, E.F.; Sheehan, D.F.; Norkett, R.; Birsa, N.; Kittler, J.T. Miro Proteins Coordinate Microtubule- and Actin-Dependent Mitochondrial Transport and Distribution. EMBO J2018, 37, 321–336, doi:10.15252/EMBJ.201696380.

Round 3
Reviewer 2 Report
The authors improved the manuscript according to my suggestions.
Some minor issues:
- Method for transmission electron microscopy is missing
- Some language errors are present